# Exploring Multivariate Profiles of Psychological Distress and Empathy in Early Adolescent Victims, Bullies, and Bystanders Involved in Cyberbullying Episodes

**DOI:** 10.3390/ijerph19169871

**Published:** 2022-08-10

**Authors:** Matteo Angelo Fabris, Claudio Longobardi, Rosalba Morese, Davide Marengo

**Affiliations:** 1Department of Psychology, University of Turin, Via Verdi 10, 10124 Turin, Italy; matteoangelo.fabris@unito.it (M.A.F.); davide.marengo@unito.it (D.M.); 2Faculty of Communication, Culture and Society, Università della Svizzera italiana, Via Buffi 13, 6900 Lugano, Switzerland; 3Faculty of Biomedical Sciences, Università della Svizzera italiana, Via Buffi 13, 6900 Lugano, Switzerland

**Keywords:** psychological distress, empathy, early adolescent victims, bullies, bystanders, cyberbullying

## Abstract

(1) Background: Adolescents may be involved in cyberbullying as victims, perpetrators, or to a lesser extent, victim–perpetrators simultaneously. The present research investigated differences between participants acting in different bullying roles—namely, bully, victim, or bully/victim—and bystander roles—namely, defending, passive bystander, and passive/defending; (2) Methods: We used multivariate analysis of covariance to determine how, in the same individuals, direct involvement in cyberbullying episodes compares to participating in them as by-standers in relation to both psychological distress and empathy; (3) Results: Both victims and bully/victims were found to be at increased risk for suicidal ideation, internalizing and externalizing symptoms, and emotional dysregulation compared with students who were neither victims nor perpetrators of cyberbullying episodes. Additionally, victims showed higher empathy scores when compared with bullies and bully/victims. All bystander roles showed increased emotional dysregulation compared with uninvolved students, but no differences emerged on other psychological distress measures. Finally, defending bystanders showed increased cognitive empathy. (4) Conclusions: During early adolescence, the direct experience of cyberbullying, as a bully or a victim (or both), show a stronger association with psychological distress than the mere participation in cyberbullying as a witness, regardless of the witness acting defensive toward the victim, or passive. However, both cyberbullying and bystanding roles provide a similar (small) explicative power over empathy variables.

## 1. Introduction

The proliferation of Internet-related technologies and social media has created increasing opportunities for forms of peer victimization [1,2,3], including cyberbullying [4,5]. It is estimated that between 10% and 60% of adolescents report experiencing cyber-victimization, while between 6% and 32% report having experienced cyberbullying, with significant geographic variation [6,7]. Adolescents may be involved in cyberbullying as victims, perpetrators, or to a lesser extent, victim–perpetrators simultaneously [8]. As with traditional bullying [9], in the online world, we find the figure of the bystander, who may be a passive observer (i.e., witnesses cyberbullying experiences but does not intervene on behalf of the victim) or an active observer (i.e., witnesses a cyberbullying episode and may intervene on behalf of the victim or in support of the bully) [10,11,12]. Faced with an episode of cyberbullying, active cyber-bystanders might intervene in favor of the victim in a variety of ways: constructively-victimizing (e.g., offering emotional support to the victim), constructively-bullying (e.g., telling the bully to stop), or aggressively (e.g., threatening the bully) [13].

Cyberbullying typically peaks in early adolescence [14], and despite estimates that it is less prevalent than traditional bullying [15,16], cyberbullying appears to have a greater impact on adolescents’ individual well-being than traditional bullying [17,18,19,20]. Therefore, considering the negative impact on individuals’ psychological adjustment, it is important to develop knowledge about the risk factors of cyberbullying in order to organize prevention and intervention programs. In this sense, socio-emotional skills, especially empathy and emotion regulation, seem to play a key role in the dynamics of cyberbullying [21,22].

Empathy is generally described as the ability to understand and share the emotional state of another and includes both cognitive and affective components that are closely related [23,24]. Overall, high levels of empathy appear to be associated with more positive developmental outcomes and more adaptive interpersonal behavior, which is associated with more prosocial behavior [24], less aggressive behavior [25], more positive peer relationships [11,26], fewer psychopathological symptoms [27,28,29], and better academic performance [30].

Emotion regulation is the ability to control one’s emotional responses; it is the ability to regulate emotions and is central to an individual’s psychological and relational well-being [31]. Difficulty regulating emotions in an adaptive manner has been linked to a variety of negative outcomes, including internalizing and externalizing symptoms, poorer academic performance, and poorer relationship quality [32,33].

Considering that socio-emotional skills are central in the psychological adaptation of individuals and in determining the quality of relationships with peers, it is likely that these constructs are central in determining the risk of involvement in cyberbullying. However, few studies have been conducted on the relationship between cyberbullying roles and dimensions relating to empathy, emotional regulation, and psychological distress.

### 1.1. Active Cyberbullying Roles, Psychological Distress, and Empathy

Active involvement in cyberbullying, both as a victim and as a perpetrator, tends to be associated with greater psychological distress and more negative developmental outcomes. Being a victim of cyberbullying tends to be associated with a range of internalized and externalized symptoms [34,35]. Perpetrators of cyberbullying also tend to report higher levels of psychological distress associated with increases in problematic behavior, hyperactivity, deviant behavior, and substance use, as well as more internalizing symptoms than non-perpetrators [35]. In terms of psychological distress, several studies show that involvement in bullying increases the risk of suicidal ideation, especially for victims and bullies/victims [20,36]. Suicide is one of the leading causes of death in adolescence [37]. Suicidal ideation tends to put individuals at risk for completed suicide and is associated with high distress and greater emotional dysregulation [38,39,40]. Those who are both victims and perpetrators of cyberbullying tend to report greater psychological distress than those who are only victims or perpetrators of cyberbullying, making this group of individuals particularly at risk for psychological adjustment.

A recent meta-analysis [41] found that cyberbullying perpetrators tend to have lower levels of empathy—both cognitive and emotional—than non-cyberbullying perpetrators. These data therefore seem to indicate a deficit in social skills among cyberbullying perpetrators, and this seems to depart from the notion that perpetrators of bullying and cyberbullying are “skilled manipulators”, as suggested in the past [42]. Regarding victims of cyberbullying, Zych and colleagues’ [41] meta-analysis found no significant relationship between cyber-victim role and empathy. However, when the construct of empathy was examined separately in its cognitive and affective components, it appeared that victims tend to report greater emotional empathy than non-victims [41]. This may be due to the fact that ongoing experiences of victimization cause victims to be more sensitive to the signals of suffering in others and, therefore, develop greater empathy for the emotional states of others. Moreover, evidence suggests that high levels of empathy may be associated with greater emotional dysregulation, which would explain the possible link between greater empathy in cyberbullying victims and increased psychological distress. However, data in the literature on the relationship between cyber-victimization and empathy are more mixed, and according to Wong and colleagues [43], these discrepancies may be due to the confounding role of the cyberbully/victim group. Indeed, some evidence suggests that cyber-victims exhibit higher levels of empathy, both cognitive and affective, compared to the cyberbully/victim group [21,44] and cyberbullies [21].

In terms of emotion regulation processes, the literature seems to suggest that both perpetrators and victims of cyberbullying tend to report greater emotional dysregulation and maladaptive emotion regulation strategies [21,45,46,47,48]. In general, it is possible that an adolescent who has difficulty recognizing and regulating his or her emotions finds on the Internet, and particularly in cyberbullying behavior, a means to channel his or her unregulated emotions toward an external subject. Along these lines, it is possible that adolescents with difficulties in emotion regulation find, in aggressive behavior, a dysfunctional coping strategy to regulate unpleasant emotional states, such as anger and frustration [10,49]. Adolescents with difficulties in emotion regulation tend to have lower social skills and tend to become more isolated and less accepted by peers, increasing the risk of cyber-victimization. Aratò and colleagues [21] have shown that victims and perpetrators of cyberbullying tend to report lower perceived social support in both family and peer groups, which reduces the opportunity to develop appropriate emotion regulation strategies and, in this way, increases the risk of becoming involved in bullying.

### 1.2. Bystander Roles in Cyberbullying Episodes, Psychological Distress and Empathy

While the literature is more developed regarding empathy, emotional regulation, and psychological distress in perpetrators, victims, and perpetrator-victims of cyberbullying, there is little information to date regarding the figure of the active or passive bystander. The bystander of a cyberbullying episode can intervene in favor of the victim, both directly (for example, by telling the cyberbullying victim to stop) and indirectly (for example, by asking an adult for help); or they can support the cyberbullying victim (for example, by liking or forwarding their posts); or ultimately remain neutral [50]. It is estimated that nearly 90% of adolescents have witnessed an episode of cyberbullying, but most of them do not intervene on behalf of the victim, usually out of concern about becoming a victim themselves or because they do not feel effective [51]. However, the cyber-bystander is not immune to mental health risks associated with involvement in bullying, although the relationship between the role of the bystander and psychological distress remains understudied. The literature seems to focus more on the reasons and factors that drive bystander actions in favor of the victim (defenders), while there is little work on the risk of mental health problems among those who observe cyberbullying. Bystanders, even in the online world, may experience a cognitive dissonance between what they should do (help the victim) and what they can do, and this dissonance could increase psychological distress [52]. In this sense, the bystander could develop negative affects related to the dissonance between believing they should intervene, but doing nothing to help the target [52,53]. Some evidence suggests that cyber-bystanders tend to report greater psychological distress compared to non-cyber-bystanders [52,54]. Consistent with the literature on traditional bullying [55], Rodelli and colleagues [36] found that cyber-bystanders were also more strongly associated with a risk of suicidal ideation than non-involved students, although no distinction was made in their study between different bystanding roles, suggesting the need for further research that provides a more nuanced understanding of this association.

Regarding empathy, few studies have addressed bystander roles in the context of cyberbullying. Several lines of evidence [41,56] found empathy to be a predictor of defense. Machackova and Pfetsch [57] found in Germany that affective empathy and cognitive empathy were independent predictors of defense. In contrast, in a sample of Polish adolescents, Barlisnka and colleagues [8] found that cognitive empathy activates prosocial cyber-bystanding behaviors, while affective empathy showed no effect. Therefore, further research needs to be conducted.

Some evidence suggests that self-regulatory abilities tend to be associated with a decline in amoral and aggressive behaviors. Along these lines, some evidence suggests that in bullying and cyberbullying episodes, self-regulatory skills tend to be more associated with constructive defense, whereas poor regulatory skills may increase the risk of aggressive defense [58,59]. In general, evidence suggests that greater self-regulation tends to promote more prosocial behaviors in bystanders, in this vein, the literature on traditional bullying suggests that active bystanders (defenders) tend to report greater social-emotional skills [58,60], including better emotion regulation [60]. Passive bystanders are more likely to intervene on behalf of the victim for a variety of reasons, including that they tend to experience more emotional distress when observing a bullying scene [61]. It is possible, therefore, that active observers may feel more able to regulate their emotions, allowing them to use the empathy they feel for bullying victims to facilitate helping behaviors [58,60,61,62]. However, more research is needed on this point, particularly with regard to the role of bystanders in cyberbullying.

### 1.3. The Present Study

Driven by previous considerations concerning the need for more research in this area, in the present study, we examine data collected in a large sample of early adolescents and look at the distribution of measures of psychological distress and empathy across groups of early adolescents reporting different roles in cyberbullying episodes. We focus on active bullying roles—namely, bully and victim—and on bystander roles—namely, passive and defending bystanders. More specifically, we look at the contributions of both active bullying and bystander roles in cyberbullying in explaining differences in multivariate profiles consisting of a comprehensive set of variables including suicidal ideation, internalizing and externalizing symptoms, emotional dysregulation, and both affective and cognitive empathy. By doing this, we aim to provide novel evidence regarding the existence of differential associations between early adolescents’ involvement in active bullying and bystander roles, and psychosocial adjustment and empathy in adolescence.

## 2. Materials and Methods

### 2.1. Sample 

The recruited sample consists of 1158 students attending middle-schools in the northwest area of Italy, with a mean age of 12.35 (SD = 0.98: range: 11–15), of which 51.4% are female, 46.7% are male, and 1.9% are non-binary students. Regarding involvement in Internet behaviors, all participants reported having used smartphone applications during the last year, including social media (e.g., Facebook, Instagram, TikTok, Menlo Park, CA, USA, ByteDance Ltd., Beijing, China etc.) or instant messaging apps (e.g., WhatsApp, Facebook’s Messenger, Telegram, etc.).

Participants were recruited in schools in northwestern Italy. After the school principal gave their consent, the study was presented to parents and students. Only those students who provided informed consent from their parents and agreed to participate in the study were recruited. There was no reward for participation in the study.

Participants were informed of the nature and objectives of the study, in compliance with the ethical code of the Italian Association for Psychology (AIP). The Institutional Review Board of the college to which the authors belong approved this procedure (n. 290961). The ethical regulations of the Italian Society of Psychology were strictly followed.

### 2.2. Instruments

#### 2.2.1. Bullying and Observer Behaviors in Cyberbullying Episodes

We administered a 16-item scale assessing four different forms of participation in cyberbullying [63]. The scales consisted of four subscales, each comprising four items; namely, cyberbullying (e.g., “I threatened or insulted someone using the Internet or the phone”), victimization (e.g., “Someone created an online group in which people made fun of me”), defending (e.g., “I defended someone who was threatened or insulted via the phone or the Internet”) and passive bystanding (e.g., “When someone was excluded from an online group of which I was a member, I minded my own business”). Items were rated using a 5-point scale: 1 (never), 2 (once during the last month), 3 (two or three times during the last month), 4 (once a week), and 5 (more than two times a week). For the purpose of the present study, we grouped participants according to their role in cyberbullying events based on responses to the aforementioned four subscales. More specifically, we created two categorical indicators, one grouping participants according to their involvement in cyberbullying episodes as a bully, victim or bully/victim, and another grouping participants based on their role when observing cyberbullying episodes; namely, defending, passive bystanding, or a more ambiguous role (both defending and passive bystanding). Participants were coded as uninvolved if they selected the 1 (Never) response to each item, while they were coded, respectively, as bully, victim, defending, or passive bystanding if they responded 2 (once during the last month) or more frequent to the respective subscale. Based on participants’ responses to the cyberbullying and victimization subscales, we grouped participants according to the following bullying roles: uninvolved students (N = 622), victim (N = 282), bully (N = 56), and bully/victim (N = 198). Based on participants’ responses to the defending and passive bystanding subscales, we grouped participants according to the following observer roles: uninvolved students (N = 370), defending (N = 258), passive bystanders (N = 88), and both defender and passive bystander (442). 

#### 2.2.2. Suicidal Ideation

We administered an Italian adaption of the Suicidal Ideation Questionnaire—Junior version (SIQ-JR), a self-report instrument which includes 15 items assessing the frequency of thoughts that adolescents who are at risk for suicide may have had during the last month (Example items: “I thought it would be better if I was not alive”, “I thought about when I would kill myself”, “I thought that no one cared if I lived or died.”). The items are rated on a 7-point scale from 0 (“I never had this thought”) to 6 (“almost every day”). The total raw scores provide an overall indicator of clinical severity within a range from 0 to 90. Higher scores indicate higher severity of suicidal ideation. In our study, the scale’s reliability was excellent (α = 0.94).

#### 2.2.3. Emotional Dysregulation

We administered an Italian adaptation of the short version of the Difficulties in Emotion Regulation Scale [64,65] that can be used to assess six aspects of emotion regulation difficulties; namely, awareness (e.g., “I pay attention to how I feel”), clarity (e.g., “I have no idea how I am feeling”), goals (e.g., “When I’m upset, I have difficulty getting work done”), impulse (e.g., “When I’m upset, I become out of control”), strategies (e.g.,” When I’m upset, I believe that I’ll end up feeling very depressed”), and non-acceptance (e.g., “When I’m upset, I feel guilty for feeling that way”). Items were rated on a 5-point Likert scale ranging from 1 (I do this almost never) to 5 (I do this almost always). The scale showed good reliability in our sample (α = 0.86).

#### 2.2.4. Externalizing and Internalizing Symptoms

Students’ internalizing and externalizing symptoms were assessed by administering a self-report instrument, the Italian version of the Strength and Difficulties Questionnaire (SDQ) [66]. Students provided a rating of their symptoms by answering 25 items that refer to the positive or negative feeling and behaviors. The items were evaluated on a 3-point Likert scale (i.e., not true, partially true, and absolutely true), and assessed five dimensions of students’ emotional and behavior characteristics: Emotional problems (e.g., “I am often unhappy, depressed or tearful ”), conduct problems (e.g., “I get very angry and often lose my temper”), hyperactivity/inattention (e.g., “I am constantly fidgeting or squirming”), peer relationship problems (e.g., “I have one good friend or more”, reversed), and prosocial behavior (e.g., “I am helpful if someone is hurt, upset or feeling ill”). Following recommendations by Goodman [66], for the purpose of this study, items were combined to create two scores reflecting students’ internalizing symptoms (sum of emotional symptoms and problematic relationships with peers), and externalizing symptoms (sum of conduct problems and hyperactivity/inattention symptoms). The Cronbach’s alpha for both components was acceptable (internalizing symptoms: α = 0.76; externalizing symptoms: α = 0.67).

#### 2.2.5. Affective and Cognitive Empathy

Empathy was measured with the Italian version of the 28-item Interpersonal Reactivity Index (IRI). The IRI includes 4 subscales, allowing for the assessment of two components of empathy: cognitive empathy (perspective taking and fantasy) and affective empathy (empathetic concern and personal distress). Example items for cognitive empathy are: “I really get involved with the feelings of the characters in a novel” (fantasy), “When I’m upset at someone, I usually try to “put myself in his shoes” for a while” (perspective taking). Example items for affective empathy are: “When I see someone being taken advantage of, I feel kind of protective towards them” (empathetic concern), “I sometimes feel helpless when I am in the middle of a very emotional situation” (personal distress). Each subscale consists of 7 items rated on a 5-point Likert scale from 1 (never) to 5 (always). The IRI has been used with early adolescent and adult participants and shows good reliability and validity (e.g., [67,68]). The Cronbach’s alpha for both affective and cognitive components was acceptable (Affective: α = 0.65; Cognitive: α = 0.72).

### 2.3. Data Analysis

We used multivariate analysis of covariance (MANCOVA) to investigate group differences in psychological distress (i.e., suicidal ideation, internalizing and external symptoms, and emotional dysregulation) and empathy (cognitive, affective) across individuals characterized by different bullying and observer roles in cyberbullying episodes. More specifically, we ran a MANCOVA model including both the indicators of bullying roles (i.e., uninvolved, bully, victim, and bully/victim) and bystander roles (i.e., uninvolved, defending, passive bystander, and defending/passive bystander) as independent (categorical) variables, as well as their interaction effect. In the analyses, we controlled for age and gender differences. In order to determine the significance of emerging multivariate differences across the groups, we refered to Wilk’s Lambda criterion, as suggested by Ateş and colleagues [69] in case of unbalanced data. Univariate F tests were inspected to determine significance of between-group differences on each variable; partial eta square (η^2^) was used to determine the effect-size of both multivariate and univariate effects. Additionally, to assess pairwise differences between groups on study outcomes, estimated marginal means for each group were compared with Bonferroni-corrected nominal *p*-values (*p* < 0.05).

Please note that to ease interpretation of parameter estimates and estimated marginal means, outcome variables were standardized prior to performing all the aforementioned analyses. Additionally, winsorizing (i.e., [70]) was used to deal with univariate outlier values (i.e., values sitting beyond an absolute Z value of 3.29 were replaced with the highest nonoutlier value). However, please note that observations including outlier values amounted to only 1.72% of data.

Because our participants were clustered in classrooms, we also expected a certain level of non-independence in the measured outcome according to the clustering of the data. For this reason, before running the analyses, we checked the intra-class correlation of the outcomes representing the correlation between individual scores due to participants belonging to the same cluster (i.e., classroom). Analyses were performed using the multilevel function of the misty package for R [71] and revealed only negligible ICCs, ranging from 0.001 (externalizing symptoms) to 0.024 (suicidal ideation). Because of these low values, we decided not to use a multilevel approach to perform the analyses; however, following recommendations by Huang [72] concerning how to deal with clustered data without multilevel analysis, in checking the significance of our results, we used design effects to adjust standard errors to control for the inflation of Type I error due to the clustering of the data. More specifically, we use the computed ICCs to compute the design effects associated with each outcome variable, assuming an average classroom size of 14; then, the square root of the design effect (i.e., DEFT) was used to correct the standard error of estimates, and finally to compute 95% confidence intervals. Except where indicated, analyses were performed using SPSS, version 23.

## 3. Results

As a first analytical step, we ran the MANCOVA model including both the indicators of bullying and observer roles, as well as their interaction effect. In this model, the interaction effect between bullying and bystander roles was not significant; for this reason, we removed the effect from the model and we will not be commenting further on it. Instead, we only discuss the model including both bullying and bystander roles as main effects. 

First, we look at between-group differences on the study measures across the four groups characterized by different active bullying roles in cyberbullying episodes. Significance of the multivariate test provided support for significant differences between bullying roles on the set of investigated variables (Wilks’s lambda = 0.84, F(18,3233.37) = 11.76, *p* < 0.001, η^2^ = 0.06). Overall, significant univariate between-group differences emerged on all variables: suicidal ideation (F(3,1148) = 29.33, *p* < 0.001, η^2^ = 0.07), internalizing symptoms (F(3,1148) = 35.92, *p* < 0.001, η^2^ = 0.09), externalizing symptoms (F(3,1148) = 42.34, *p* < 0.001, η^2^ = 0.10), emotional dysregulation (F(3,1148) = 16.16, *p* < 0.001, η^2^ = 0.04), affective empathy (F(3,1148) = 3.34, *p* < 0.05, η^2^ = 0.01), and cognitive empathy (F(3,1148) = 4.62, *p* < 0.01, η^2^ = 0.01). 

Based on Bonferroni-corrected pairwise contrasts, the four groups showed different patterns on the study measures. Between-group differences are visualized in Figure 1. In the figure, groups showing significant mean differences in the study outcomes are labeled with different letters. 

First, it emerged that both uninvolved students and bullies reported significantly lower scores on suicidal ideation and internalizing symptoms when compared with students categorized as victims and bully/victims. Regarding externalizing symptoms, uninvolved students reported the lowest score when compared with all the other groups, bully/victims showed the highest mean scores, while bully and victim students showed no differences and shared a similar intermediate position on this scale. Uninvolved students also showed the lowest mean scores on emotional dysregulation when compared with victim and bully/victim students, while bully students showed no significant differences when compared with the other groups. Finally, regarding empathy, victims showed significantly higher scores than uninvolved students on the affective empathy component, while the other groups showed no significant differences; in turn, both victims and uninvolved students scored higher on the cognitive empathy measure than bully/victim students, which showed the lowest scores.

We also look at between-group differences on the study measures across the four groups characterized by different bystander roles in cyberbullying episodes. Again, significance of the multivariate test provided support for significant differences between these groups on the set of investigated variables (Wilks’s lambda = 0.96, F(18,3233.37) = 2.39, *p* < 0.01, η^2^ = 0.01). Between-group differences are visualized in Figure 2. 

Overall, significant univariate between-group differences emerged only on two variables; namely, emotional dysregulation and cognitive empathy: suicidal ideation (F(3,1148) = 0.64, *p* = 0.59, η^2^ = 0.00), internalizing symptoms (F(3,1148) = 0.15, *p* = 0.93, η^2^ = 0.00), externalizing symptoms (F(3,1148) = 23.50, *p* < 0.01, η^2^ = 0.06), emotional dysregulation (F(3,1151) = 22.56, *p* < 0.01, η^2^ = 0.06), affective empathy (F(3,1148) = 3.01, *p* < 0.05, η^2^ = 0.01), and cognitive empathy (F(3,1148) = 6.57, *p* < 0.01, η^2^ = 0.02). 

In Figure 2, we labeled groups, showing significant mean differences on the study outcomes with different letters; please note that letter labels are not displayed when outcomes did not show significant between-group differences. As noted above, between-group differences were only detected on emotional dysregulation and cognitive empathy. In more detail, it emerged that all groups reporting some form of involvement as a bystander in cyberbullying episodes (i.e., passive, defending, and passive/defending) showed higher scores on emotional dysregulation than uninvolved students. Finally, defending bystanders showed higher cognitive empathy scores than all the other bystander groups and uninvolved studies, which in turn showed no significant difference between group differences.

Looking at results emerging from both main effects (i.e., bullying roles and bystander roles) it appears that when these are included together in the same model, individual differences in bullying roles provide a stronger contribution than bystander roles in explaining differences in the multivariate profile. In more detail, the variability of psychological distress measures can be traced back prevalently to individual differences in bullying roles, as opposed to differences in bystander roles. However, both bullying and bystander roles provide similar (small) contributions in explaining the variability of empathy measures.

## 4. Discussion

In the present study, we investigated differences in measures of psychological distress and empathy across groups of students characterized by different levels of involvement in cyberbullying episodes. More specifically, we looked at differences between participants acting in different bullying roles—namely, bully, victim, or bully/victim—and bystander roles—namely, defending, passive bystander, and passive/defending. Using multivariate analysis, we were able to determine how, in the same individuals, direct involvement in cyberbullying episodes compares to participating in them as bystanders in relation to both psychological distress and empathy.

Overall, both victims and bully/victims were found to be at an increased risk for suicidal ideation, internalizing and externalizing symptoms, and emotional dysregulation compared with students who were neither victims nor perpetrators of cyberbullying episodes. These data are consistent with previous literature suggesting that involvement in bullying increases risk for mental health problems, including internalizing and externalizing symptoms, suicidal ideation, and emotional dysregulation. Cyberbullying often consists of aggressive actions that ridicule, humiliate, and devalue the image of the person being bullied. The literature indicates an increased risk of suicide among individuals involved in cyberbullying, particularly victims and bully/victims [20,36]. In addition, there is evidence that cyberbullying is associated with a greater increase in suicide risk compared to traditional bullying, likely due to the way cyberbullying is characterized [73]. The literature on bullying [74] and cyberbullying [75,76], as well as peer victimization in general [77,78], tends to report an increased risk of internalizing symptoms among victims; moreover, consistent with previous literature, the role of the bully/victim emerges as a particularly critical element, taking on the role of both victim and aggressor. It is possible that the bully/victim engages in aggressive behaviors as a strategy to channel his or her impulses, which in turn, puts him or her at risk of more negative interactions with peers and a sense of isolation from the group, increasing the risk of victimization. Consistent with the cyclical process model [46] it is also possible that a victimized student may find a revenge mode in the role of cyberbully, which triggers a barren cycle that exposes the subject to increased psychological distress. According to this model, a victimized adolescent who develops feelings of anger and frustration turns to media with antisocial and risky content, which subsequently reinforces the performance of cyberbullying acts. In this way, cyberbullying victims often become victims again, and enter a cyclical loop that leads to becoming a victim, becoming a cyberbully, and being bullied again. Cyberbullying victims tend to be at a higher risk for externalized disorders, likely reflecting their difficulties to form more positive relationships with others and regulate their impulses. These difficulties increase the risk of aggressive behaviors, exposing them to rejection and isolation by their peer group. Typically, adolescents with externalizing symptoms tend to be rejected by peers [79] and have low social status among peers, which increases their risk of being targeted for aggressive acts [80]. It is also possible that individuals with low social skills are victims of aggressive acts in the peer group and that they find some kind of revenge on the Internet, thus, becoming aggressors in the online world. However, it is possible that difficulties in social skills are also characteristic of online interactions and that the use of social media and cyberspace exposes such individuals to additional risks of victimization online.

Regarding empathy, cyber-victims tend to show greater affective empathy than individuals who are not involved in cyberbullying. This is consistent with some evidence suggesting that cyberbullying victims show greater affective empathy, likely as a result of victimization experiences that cause them to be more attentive and sensitive to the signs of distress in others [41]. However, our data do not suggest significant differences between other cyberbullying roles in terms of affective empathy. With regard to cognitive empathy, our data suggest that uninvolved students and cyber-victims tend to report higher scores than cyberbullying perpetrators, particularly bullies/victims, which is consistent with literature identifying cyberbullies/victims [21] as the group with greater empathic deficits. Overall, these data seem to suggest that victims of cyberbullying tend to retain better empathic skills than the aggressors, who thus, do not appear to be skilled manipulators as described in the past, but rather exhibit deficits in their ability to recognize and share the feelings of others.

As for the role of the cyber-bystander, the literature on this topic is underdeveloped, as noted earlier, and further research is needed. Overall, our data seem to suggest that individuals who have been involved in some form of cyberbullying tend to report greater emotional dysregulation than those who have never witnessed cyberbullying incidents, but no differences were detected regarding the other measures of psychological distress. Additionally, our data do not seem to support the idea that cyber-defenders report better psychological well-being and a greater ability to regulate emotions than other cyber-defenders. Indeed, it has been suggested that better social-emotional skills and the ability to regulate emotions are factors that promote greater defense activation [58,60,61,62]. In this sense, greater coping skills in stressful situations would have made defenders more confident in their abilities to intervene on behalf of the victim. However, our data do not show significant differences between the roles of defenders and other cyber-bystanders on measures of emotion regulation. It is possible that intervening on behalf of the cyber-victim is more related to other factors, such as empathy. Along these lines, our data revealed that cyber-defenders had higher scores on cognitive empathy than other bystander roles, while there was no difference between bystander groups on affective empathy. These data are consistent with the results of a study with Polish adolescents [8], which found that cognitive empathy activated prosocial cyber-defense behavior, whereas affective empathy showed no effect. However, the role of empathy, both affective and cognitive, in promoting helping behaviors in the cyber-bystander, has yielded mixed results in the literature to date, and we have very little knowledge about the figure of the cyber-bystander. Further research is needed in the future. 

Overall, findings from the present study indicate that when cyberbullying roles and cyber-bystanding roles are examined together in a single analysis, cyberbullying roles appear to explain most of the variability in psychological distress, while both cyberbullying and cyber-bystanding roles provide a similar (small) explicative power over empathy variables. This indicates that during early adolescence, the direct experience of cyberbullying as a bully or a victim (or both), is expected to show a stronger association with psychological distress than the mere participation in cyberbullying as a witness, regardless of the witness acting defensively or passively toward the victim. This novel result suggests caution in interpreting significant associations between cyber-bystanding behavior and psychological distress when information about direct involvement in cyberbullying as a bully or victim is not controlled for in the model (see for example [52]).

### 4.1. Limits and Future Direction

Overall, our study seeks to expand our current knowledge of possible differences between cyberbullying activists on the one hand, and bystanders on the other, with respect to dimensions of psychological distress and empathy. Few studies have been conducted along these lines that consider these variables and roles in a single study. Therefore, the data cannot be considered conclusive, and further research is needed. The results of our study must be understood in light of the various methodological limitations of the study. Indeed, our sample is limited in number and was recruited through a few schools in northwestern Italy. This severely limits the generalizability of the results to the target population. Future studies could, therefore, include larger and more representative samples, and cross-cultural comparisons are also recommended to identify the possible influence of cultural factors on the relationship between the variables studied. In addition, our study is a cross-sectional study, so it is not possible to make a statement about the direction of causality of the relationship between the variables studied. Therefore, future studies can be replicated by relying on longitudinal studies. The research protocol consisted of a self-report questionnaire. Therefore, factors such as social desirability, memory, and text comprehension may have influenced subjects’ responses. Future studies could, therefore, rely on external observers (e.g., teachers, psychologists, or other data sources) and other research methods.

### 4.2. Practical Implications

Our data seem to offer some practical implications for prevention and intervention. Involvement in cyberbullying, in whatever role, tends to increase the risk of psychological distress, especially for victims and victims of bullying. Therefore, cyberbullying prevention and response measures are important for both those involved in an active and passive role to prevent forms of psychopathology, including the risk of suicidal ideation. Intervention and prevention programs must take into account the role that the youth play in bullying. Promoting empathy skills in adolescents, especially cognitive skills, could be a strategy to reduce bullying by stimulating defensive actions in the peer group.

## 5. Conclusions

In summary, our study seeks to explore possible differences within two groups of cyberbullying roles: those who actively intervene as victims, perpetrators, or both (cyberbullying/victims), and the cyberbullying roles. Our data show some differences between the two groups in factors such as psychological distress and empathy. Specifically, involvement in cyberbullying tends to predict greater psychological distress, including internalizing and externalizing symptoms, and suicidal ideation. In addition, cyberbullies/victims tend to report greater psychological distress than other groups, while cyber-victims show higher levels of affective empathy than those not involved in bullying and greater cognitive empathy than other cyber-roles. The data suggest that those who are involved in cyberbullying as observers tend to report greater emotional dysregulation than those who are not involved. Finally, it appears that cognitive empathy is a characteristic that distinguishes cyber-defenders from cyberbullying roles.

## Figures and Tables

**Figure 1 ijerph-19-09871-f001:**
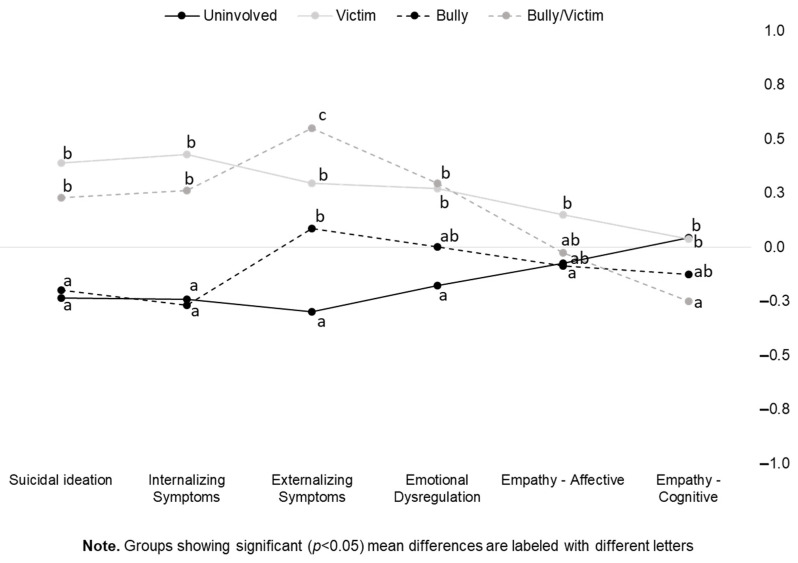
Multivariate profile of psychological distress and empathy by a bullying role in cyberbullying episodes.

**Figure 2 ijerph-19-09871-f002:**
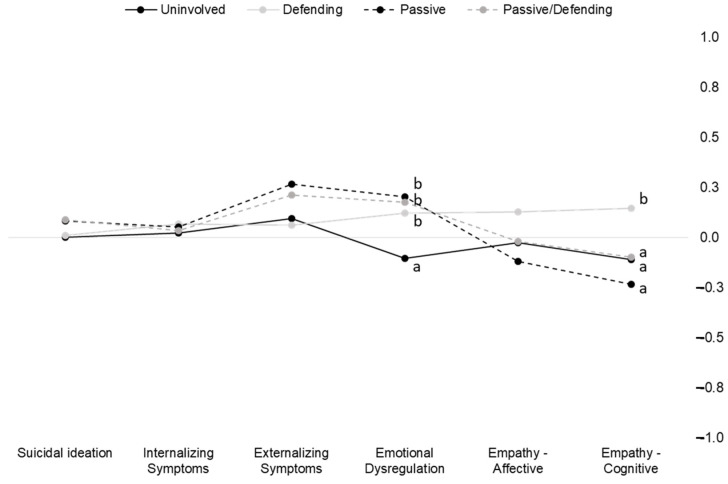
Multivariate profile of psychological distress and empathy by a bystander role in cyberbullying episodes.

## Data Availability

Not applicable.

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
