# Peer review of "Exploring Multivariate Profiles of Psychological Distress and Empathy in Early Adolescent Victims, Bullies, and Bystanders Involved in Cyberbullying Episodes"

_ijerph, 2022, doi:10.3390/ijerph19169871_

Round 1

Reviewer 1 Report

The present study is focused on cyberbullying among adolescents, especially on differences between particular profiles created by degree of involvement in cyberbullying in relation to psychological distress and empathy.  Topic of cyberbullying is currently highly discussed and examined as digital technologies became important part of adolescents’ lives. Generally, this article has a good quality and contribute to existing knowledge, however small issues should be improved.

Based on background information, authors provided good knowledge of this topic. Nevertheless, introduction should be shortened a condensed. For example, avoid to provide definitions, especially definition of concepts which are generally known, like empathy. Moreover, association between cyberbullying and suicidal ideation should be assessed more explicitly. I miss the reasoning why authors are focused on this relationship.

The basic characteristics of sample should be indicated in table (gender, age, bullying roles), for that reason part of instrument description P5 L236-242 might be skipped and moved to this table.

Author Response

The present study is focused on cyberbullying among adolescents, especially on differences between particular profiles created by degree of involvement in cyberbullying in relation to psychological distress and empathy.  Topic of cyberbullying is currently highly discussed and examined as digital technologies became important part of adolescents’ lives. Generally, this article has a good quality and contribute to existing knowledge, however small issues should be improved.

Based on background information, authors provided good knowledge of this topic. Nevertheless, introduction should be shortened a condensed. For example, avoid to provide definitions, especially definition of concepts which are generally known, like empathy. Moreover, association between cyberbullying and suicidal ideation should be assessed more explicitly. I miss the reasoning why authors are focused on this relationship.

We thank you for your suggestions. We have tried to reduce the text in some parts and deepen the points you indicated.

The basic characteristics of sample should be indicated in table (gender, age, bullying roles), for that reason part of instrument description P5 L236-242 might be skipped and moved to this table.

Thank you for your suggestion. We believe adding a table to the manuscript would not improve the readability of the manuscript. For this reason, we have decided to keep this information in the text.

Reviewer 2 Report

First of all, congratulations to the authors for the article.  It is very interesting and timely, along the lines of so many other articles on the subject in question.

I consider that the introduction is quite extensive, perhaps justified by the complexity of the subject; however, I believe that presenting the concepts and the current state of the art would be sufficient.

The design is adequate although, from my point of view, with important limitations as the authors themselves reflect in the text.  I think it would have been richer to develop a mixed research, using questionnaires with focus groups and interviews with parents, teachers and peer groups.  Undoubtedly, it would have been more interesting and conclusive.

Otherwise, I find it acceptable.

Best regards.

Author Response

First of all, congratulations to the authors for the article.  It is very interesting and timely, along the lines of so many other articles on the subject in question.

I consider that the introduction is quite extensive, perhaps justified by the complexity of the subject; however, I believe that presenting the concepts and the current state of the art would be sufficient.

The design is adequate although, from my point of view, with important limitations as the authors themselves reflect in the text.  I think it would have been richer to develop a mixed research, using questionnaires with focus groups and interviews with parents, teachers and peer groups.  Undoubtedly, it would have been more interesting and conclusive.

Otherwise, I find it acceptable.

Best regards.

 Thank you very much for your comment and rating. We have made an effort to shorten the introduction. However, you are correct that it is difficult to shorten the text due to the complexity of the relationships between the variables under study.

Reviewer 3 Report

This is a review of the manuscript titled "Exploring multivariate profiles of psychological distress and empathy in early-adolescent victims, bullies, and bystanders involved in cyberbullying episodes". This study aims to examine the effects of bullying (bully, victim, bully/victim, and uninvolved) and bystander (defending, passive, passive/defending, and uninvolved) roles. The results showed that suicidal ideation, emotional dysregulation, internalizing symptoms and externalizing symptoms were higher in victims and perpetrators/victims. This manuscript could be improved if the following concerns are addressed:

1. It is stated that cyberbully is defined as "repeated, intentional aggression carried out through an electronic medium against a victim who is less able to fight back" (p. 1). However, this study also proposes that a person can be a bully and victim at the same time. Moreover, the literature on bullying suggests that there are aggressive victims. It is questionable that a victim is less able to fight back.

2. The authors state that "there is no agreed-upon definition of cyberbullying among experts, and heterogeneity in measurement tools makes it difficult to estimate the prevalence of the phenomenon among youth" (p. 1). Please provide more details about the various definitions and measures of cyberbullying.

3. The authors state that "victims tend to report greater emotional empathy than non-victims". Has there been any research evidence? Please provide references for this claim.

4. It is hypothesized that victims have greater empathy but poorer emotion regulation. However, empathy and emotion regulation are highly associated with each other. Why a victim with poor emotion regulation can have a high level of empathy?

5. The authors state that bystanders "may experience a cognitive dissonance between what they should do (help the victim) and what they can do, and this dissonance could increase psychological distress" (p. 4). Are there any references for this point? Please explain in more detail how the experience of a bystander can lead to cognitive dissonance.

6. Please provide the reference for the Suicidal Ideation Questionnaire – JR version (SIQ-IR) (p. 5). Please explain what "JR" stands for.

7. The full form of MANCOVA is not "multivariate analyses of variance with covariates" (p. 6)

8. The authors conducted two separate MANCOVAs to examine the effects of bullying (bully, victim, bully/victim, and uninvolved) and bystander (defending, passive, passive/defending, and uninvolved) roles. These two factors should be examined simultaneously in one two-way MANCOVA. The two-way MANCOVA should not only use for examining effect sizes but also the mean differences. Besides, the interaction effects should also be reported.

Author Response

This is a review of the manuscript titled "Exploring multivariate profiles of psychological distress and empathy in early-adolescent victims, bullies, and bystanders involved in cyberbullying episodes". This study aims to examine the effects of bullying (bully, victim, bully/victim, and uninvolved) and bystander (defending, passive, passive/defending, and uninvolved) roles. The results showed that suicidal ideation, emotional dysregulation, internalizing symptoms and externalizing symptoms were higher in victims and perpetrators/victims. This manuscript could be improved if the following concerns are addressed:

  1. It is stated that cyberbully is defined as "repeated, intentional aggression carried out through an electronic medium against a victim who is less able to fight back" (p. 1). However, this study also proposes that a person can be a bully and victim at the same time. Moreover, the literature on bullying suggests that there are aggressive victims. It is questionable that a victim is less able to fight back.

You are right. We thank you for your contribution. In line with the other reviewers, we deleted the definition of cyberbullying. We will keep the descriptions of roles in cyberbullying in the text.

  1. The authors state that "there is no agreed-upon definition of cyberbullying among experts, and heterogeneity in measurement tools makes it difficult to estimate the prevalence of the phenomenon among youth" (p. 1). Please provide more details about the various definitions and measures of cyberbullying.

Thank you for your comment. We have eliminated this sentence by meeting the reviewers' requests to reduce the introduction as much as possible. 

  1. The authors state that "victims tend to report greater emotional empathy than non-victims". Has there been any research evidence? Please provide references for this claim.

We have added references.

  1. It is hypothesized that victims have greater empathy but poorer emotion regulation. However, empathy and emotion regulation are highly associated with each other. Why a victim with poor emotion regulation can have a high level of empathy?

The request to reduce the introduction does not allow us to deepen. However, we have added a comment on the point you indicated. We thank you.

  1. The authors state that bystanders "may experience a cognitive dissonance between what they should do (help the victim) and what they can do, and this dissonance could increase psychological distress" (p. 4). Are there any references for this point? Please explain in more detail how the experience of a bystander can lead to cognitive dissonance.

Ok. We have made the changes indicated. 

  1. Please provide the reference for the Suicidal Ideation Questionnaire – JR version (SIQ-IR) (p. 5). Please explain what "JR" stands for.

Ok. We proceeded. "JR" stands for Junior.

  1. The full form of MANCOVA is not "multivariate analyses of variance with covariates" (p. 6)

 Thank you for pointing out this error. We corrected it.

  1. The authors conducted two separate MANCOVAs to examine the effects of bullying (bully, victim, bully/victim, and uninvolved) and bystander (defending, passive, passive/defending, and uninvolved) roles. These two factors should be examined simultaneously in one two-way MANCOVA. The two-way MANCOVA should not only use for examining effect sizes but also the mean differences. Besides, the interaction effects should also be reported.

We agree with the reviewer suggestion, and revised the document accordingly. We edited the sections “2.3. Data analysis” and “3.3. Multivariate profiles of psychological distress and empathy: comparing the contribution of bullying and bystander roles”, providing additional information about significance of the interaction effect and of pairwise differences emerging from the model including both bullying and bystander roles variables.

Round 2

Reviewer 3 Report

This is a review of the revised manuscript titled "Exploring multivariate profiles of psychological distress and empathy in early-adolescent victims, bullies, and bystanders involved in cyberbullying episodes". The manuscript has been improved. There is one remaining issue:

Regarding the main effects of bullying (bully, victim, bully/victim, and uninvolved) and bystander (defending, passive, passive/defending, and uninvolved) roles, the authors should not conduct two separate MANCOVAs. The results of the main effects and post-hoc tests of the two-way MANCOVA should be reported.

Author Response

Reviewer #3’s comment:

This is a review of the revised manuscript titled "Exploring multivariate profiles of psychological distress and empathy in early-adolescent victims, bullies, and bystanders involved in cyberbullying episodes". The manuscript has been improved. There is one remaining issue:

Regarding the main effects of bullying (bully, victim, bully/victim, and uninvolved) and bystander (defending, passive, passive/defending, and uninvolved) roles, the authors should not conduct two separate MANCOVAs. The results of the main effects and post-hoc tests of the two-way MANCOVA should be reported.

Response

We acknowledge the reviewer’s suggestion to report only the results of a single MANCOVA model including the results of main effects and post-hoc tests. We have now revised the manuscript according to this suggestion. More in details, we have revised the “The present study”, rephrasing the aims of the study. We have also introduced changes in the “Data analysis” section, and “Results” section. More in details, now we only refer to single MANCOVA model including both main effects, namely bullying (bully, victim, bully/victim, and uninvolved) and bystander roles (defending, passive, passive/defending, and uninvolved). Because of the changes introduce in the data analysis, numerical results and figures were updated, and the “Results” selection no longer includes different subsections; Table 1 was also removed as it no longer needed. Finally, we acknowledged these changes in emerging results also in the “Discussion” and “Conclusions” sections, which were slightly edited. Minor changes were also introduced in the abstract.